# Navigation Engine Design for Automated Driving Using INS/GNSS/3D LiDAR-SLAM and Integrity Assessment

**Kai-Wei Chiang [1], Guang-Je Tsai [1,*] , Yu-Hua Li [1], You Li [2] and Naser El-Sheimy [2]**

[1]  Departement of Geomatics Engineering, National Cheng-Kung University, No. 1, Daxue Road, East District, Tainan City 701, Taiwan; kwchiang@mail.ncku.edu.tw (K.-W.C.); p68001013@ncku.edu.tw (Y.-H.L.)
[2]  Departement of Geomatics Engineering, The University of Calgary, 2500 University Dr NW, Calgary, AB T2N 1N4, Canada; li29@ucalgary.ca (Y.L.); elsheimy@ucalgary.ca (N.E.-S.)
[*]  Correspondence: p68031050@ncku.edu.tw

**Abstract:** Automated driving has made considerable progress recently. The multisensor fusion system is a game changer in making self-driving cars possible. In the near future, multisensor fusion will be necessary to meet the high accuracy needs of automated driving systems. This paper proposes a multisensor fusion design, including an inertial navigation system (INS), a global navigation satellite system (GNSS), and light detection and ranging (LiDAR), to implement 3D simultaneous localization and mapping (INS/GNSS/3D LiDAR-SLAM). The proposed fusion structure enhances the conventional INS/GNSS/odometer by compensating for individual drawbacks such as INS-drift and error-contaminated GNSS. First, a highly integrated INS-aiding LiDAR-SLAM is presented to improve the performance and increase the robustness to adjust to varied environments using the reliable initial values from the INS. Second, the proposed fault detection exclusion (FDE) contributes SLAM to eliminate the failure solutions such as local solution or the divergence of algorithm. Third, the SLAM position velocity acceleration (PVA) model is used to deal with the high dynamic movement. Finally, an integrity assessment benefits the central fusion filter to avoid failure measurements into the update process based on the information from INS-aiding SLAM, which increases the reliability and accuracy. Consequently, our proposed multisensor design can deal with various situations such as long-term GNSS outage, deep urban areas, and highways. The results show that the proposed method can achieve an accuracy of under 1 meter in challenging scenarios, which has the potential to contribute the autonomous system.

**Keywords:** inertial navigation system and global navigation satellite system (INS/GNSS); light detection and ranging (LiDAR); simultaneous localization and mapping (SLAM)

## 1. Introduction

Vehicular technology has gained increasing popularity recently, of which autonomous driving is a hot topic. To achieve safe and reliable intelligent transportation systems, accurate positioning technologies need to be built. In terms of safety, positioning technologies for automated vehicles are essential [1–3]. Kuutti et al. [4] provided a general picture of the recently developed conventional positioning technologies. Generally, the Global Navigation Satellite System (GNSS) is the most popular positioning system and almost everyone can easily access its service by only using his or her phone. GNSS is a satellite-based navigation system involving different constellations. It provides nearly uniform and absolute positioning determination, allowing the user to conduct positioning on land, in sea, and even in space [5,6]. However, the accuracy of GNSS varies according to

different environments. It must be integrated with other navigation systems to meet the demand of higher-accuracy applications. Traditional positioning algorithms used for automated vehicles employ an integrated system combining an inertial navigation system (INS) and GNSS [7–10]. The inertial measurement unit (IMU) is a crucial component to the INS, including at least three axes of accelerometers and gyroscopes. Its observations (specific forces and angular rates) are able to continuously track the position and orientation of a vehicle from a known initial point. However, its navigation performance drops quickly over time in accordance with the grade of the IMU itself. In addition, INS operation is based on the relative positioning mechanics, which requires the initial information (position, velocity, and orientation) to operate properly. On the other hand, GNSS provides the absolute position and velocity. The accuracy of these measurements is highly dependent on the environment [11]. In summary, INS and GNSS both have advantages and disadvantages, but their complementary characteristics have made them one of the most popular positioning systems.

The complementary properties of INS/GNSS benefit the individual limitations of each part and make it an optimal integration system for most automated use. However, such an integrated system also suffers from GNSS outage and multipath interference or non-line-of-sight (NLOS) reception, particularly in a deep urban area [12,13]. Several publications with respect to enhancing INS/GNSS have proposed solutions over the last decade, particularly for an integrated system with low-cost MEMS-based inertial sensors [14–16]. Besides the approaches to improve the integrated model itself, utilizing the other external and complementary sensors also plays a significant role in an integrated navigation system. For land-vehicle applications, vehicle measurements must be considered in the integration system, such as the velocity measured by the odometer [14,17] or the steering angle obtained from the electrical control unit [18]. Other research areas referring to integration with external sensors involve the use of barometers (to control the height drift) [19,20], magnetometers (to mitigate the heading drift), and wireless sensors [16,21,22].

In addition to positioning sensors, perception sensors have also begun to be utilized as a navigation solution. Simultaneous localization and mapping (SLAM) has been broadly researched. The perception sensors, such as ultrasonic, radar, camera, or light detection and ranging (LiDAR), were originally deployed for monitoring or mapping the environment. With the widespread development of the SLAM algorithm, they can conduct the positioning and mapping of the environment simultaneously. The basic concept of SLAM is to estimate relative transformation based on static objects or features surrounded by sensors. For example, radar and ultrasonic SLAM are relatively less sensitive to the weather condition and relatively inexpensive and low power-consuming. Radar-SLAM can act as the odometry to provide the velocity information [23] or conduct localization using the map information [24,25]. Visual-SLAM uses a monocular or stereo camera to track the features of consecutive images while estimating the relative orientation and translation [26–29]. LiDAR has the advantages of a larger measurement range and higher robustness to environment changes. The iterative closest point (ICP) is a 6-DOF LiDAR-SLAM registration method that does not use feature information [30]. This method is suitable for dense point clouds and matches only the closest points. Feature-based LiDAR-SLAM, such as LiDAR odometry and mapping (LOAM), extracts the features in a consecutive point cloud [31]. Based on the matching of fewer key features, the feature-based method has a relatively low power consumption and can achieve real-time applications.

Precise relative positioning allows SLAM to continuously localize itself and can further improve the performance when detecting loop closure on the map [32]. However, the relative positioning algorithm still suffers from an increase in accumulation error with the traveled distance. Moreover, once the solution of SLAM diverges, it is difficult for it to relocalize itself at the correct position. Thus, multisensor integration is necessary to achieve a more robust and accurate positioning system for automated vehicles. By integrating INS into SLAM, a trustable initial value for reducing the risk in divergence or loop closure problems can be achieved [33]. Meanwhile, the absolute position and velocity from GNSS also benefit SLAM in mitigating the accumulating error [34]. On the other hand, the navigation information from SLAM acts as the external source into INS/GNSS to enhance the

overall performance, even in a GNSS-denied environment [35]. With assistance from a prior map, the integrated positioning systems with LiDAR-SLAM provide a reliable and accurate positioning solution [36,37].

The existing approaches have been shown to be effective in small-scale indoor and GNSS-friendly areas; however, most of the previous publications lack testing data in a deep urban area. The most challenging problem is not the blocking of GNSS signals. Instead, multipath interference or NLOS reception are the main factors that result in a worse positioning solution. Furthermore, the height accuracy is less discussed. Having accurate height information is necessary to fully achieve automated vehicles, especially in multilayer highways, underground parking lots, and even for vehicular power control.

For meeting the higher-accuracy demand, this paper proposes a novel fusion design using 3D LiDAR-SLAM to provide more accurate measurements (velocity and heading). To acquire more information from SLAM, 3D SLAM is adopted and modified to be customized in the fusion scheme. Figure 1 illustrates the flowchart of an INS/GNSS/3D LiDAR-SLAM integrated navigation algorithm, where the following three significant contributions are proposed. INS/GNSS with vehicle motion constraints is a conventional positioning method for obtaining the navigation solution. With the concept of the integrated system, the proposed INS-aiding 3D LiDAR SLAM (green part) uses the information from INS and generates measurements through a 3D LiDAR-SLAM extended Kalman filter (EKF), feeding into the central EKF. The designed structure continuously takes the feedback information (biases and scales) to adjust INS to mitigate the drift error. Instead of directly using the measurements from LiDAR-SLAM, the SLAM position velocity acceleration (PVA) model is used to deal with the high dynamic movement. Finally, the integrity assessment ensures the reliability of each measurement. This process can be further applied to use velocity for detecting the vehicle motion and applying motion constraints such as nonholonomic constraint (NHC), zero velocity update (ZUPT), and zero integrated heading rate (ZIHR). Moreover, it also benefits the SLAM to recognize the divergence. In general, there are three major proposed algorithms in this paper—the INS-aiding 3D LiDAR SLAM, the SLAM-PVA model and the integrity assessment. The major description of the proposed methods is in Section 2.3. The following sections will provide the details accordingly.

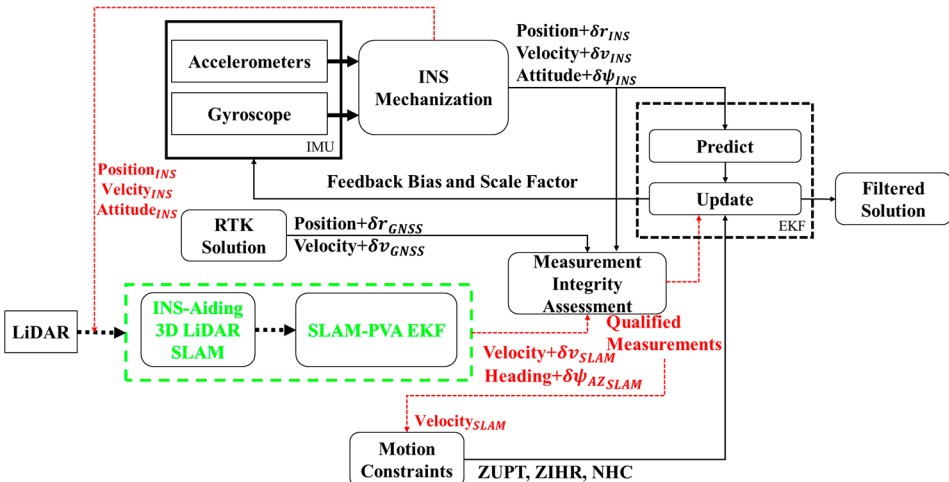

**Figure 1.** Flowchart of the inertial navigation system (INS)/ global navigation satellite system (GNSS)/3D light detection and ranging-simultaneous localization and mapping (LiDAR-SLAM) fusion design.

## 2. Methodology

### 2.1. INS/GNSS with Land Vehicle Motion Constraints

Generally, using an EKF model for INS/GNSS leads to stable results and provides robust performance in most situations. In this study, the INS/GNSS scheme is implemented, which follows the implementation in [14]. The state vector in EKF includes 21 states to accurately estimate the navigation solution, which is expressed as:

$$x_k = \begin{bmatrix} r & v & \psi & b_a & b_g & s_a & s_g \end{bmatrix}^T_{21\times1},\tag{1}$$

$$\hat{x}_k = x_k + \delta x_k,\tag{2}$$

where subscript $k$ is the parameter at time $k$; $x$ is the state vector; $\delta x$ is the error of the state vector; $\hat{x}_k$ is the updated state that is corrected by the estimated states; $r$, $v$, and $\psi$ are the major navigation information representing position, velocity, and attitude, respectively; and $b$ and $s$ represent the biases and scale factors of the accelerometers and gyroscopes, respectively.

In terms of the integrated navigation system, the system model for inertial data integration is a highly nonlinear case. Additionally, the current states are determined by integrating IMU observations using the mechanization equations. To implement the integrated INS/GNSS navigation system, it is necessary to linearize the system model to satisfy the KF assumptions. The system model of EKF is in a discrete-time form for estimating the state errors ($\delta x$) propagating in time instead of estimating the navigation states:

$$\delta x_{k+1} = \Phi_{k,k+1}\delta x_k + w_k,\tag{3}$$

where $\Phi_{k,k+1}$ represents the state transition matrix and $w_k$ represents the white noise sequence of the system. This paper adopts the psi-angle error model, especially for low-cost inertial navigation. For more details, refer to [14].

Most measurement equations in the integrated navigation system are nonlinear, such as GNSS and vehicle velocity measurement models; thus, it is necessary to linearize the equation, which can be expressed as:

$$\delta z_k = H_k\delta x_k + e_k,\tag{4}$$

where $H_k$ is a design matrix that maps the states into measurements and $e_k$ is the white noise sequence of the measurements. For details on different measurement models, such as GNSS and vehicle-frame velocity measurement models, refer to [14].

In general, GNSS is intrinsically linked with INS because of its complementary characteristics, such as the absolute positioning solution and time-irrelevant accuracy. Nevertheless, the GNSS solution still suffers from the signal block and multipath NLOS reception. However, there are several vehicle motion constraints according to the physical behavior of the automobile which contribute to the navigation system as the pseudo-measurements (pseudo-measurements are acquired by an assumption, not from the sensors) and mitigate the impact of the failure of the GNSS solution to some extent.

The first constraint is to limit velocity in the lateral and vertical directions, which is called NHC. According to the standard physical behavior of moving land vehicles, there should be no jumps or lateral slides. In other words, if the vehicle is moving, NHC limits the lateral and vertical velocities of the vehicle to be zero as the pseudo-measurement into EKF [14,16]. The second constraint involves two updates in velocity and attitude domains, ZUPT and ZIHR. These two pseudo-measurements assume that the velocity and integrated heading are zero as the static state is detected. By using these two updates, the accumulating error in velocity and heading can be effectively suppressed and controlled. In addition, the stationary situation often occurs in the urban area, as the vehicle needs to stop and start frequently.

## 2.2. Feature-Based 3D LiDAR SLAM (LiDAR Odometry and Mapping)

In general, SLAM can be simplified into two models: motion and measurement models [32]. Based on this principle, the 3D LiDAR-SLAM used in this study can also be divided into LiDAR odometry and LiDAR mapping. Moreover, feature extraction plays a pivotal role in finding the corresponding feature in each scan (the total point cloud measurements within a 360° horizontal rotation). The concept of these three processes is based on LOAM [31].

### 2.2.1. Feature Extraction

For the registration process in LiDAR odometry, it is inefficient and not necessary to align all the point clouds simultaneously. To efficiently increase the speed of calculation and maintain the positioning performance, the significant features are extracted based on the geometry in the point cloud. In the feature extraction process, the edge point (point on the edge of the object) and planar point (point on the planar surface) are extracted with a co-planar geometric relationship from individual channels. In terms of the Velodyne LiDAR (VLP-16) used in this study, there are 16 channels for an individual scan. Each channel has equally sized scan regions and is divided into 6 subregions (S) for evenly distributing the features within the environment. The definition of curvature ($c$) for determining the edge and planar points is written as follows [31]:

$$c = \sum_{j \in S, j \neq i}^{w} \left( p_{k,i}^{l} - p_{k,j}^{l} \right)^2,$$

(5)

where $c$ is the curvature for evaluating the smoothness of the local surface, w is the total number of points in the subregion, $i$ is a point in the subregion, and $j$ is the consecutive point of $i$-th point.

The curvature of each point is sorted according to its value. If the curvature is higher than the preset threshold, it is determined as the edge point, and vice versa. However, limited features are extracted in each subregion. For each subregion, a maximum of two edge points and four planar points can be selected.

In addition, two situations might lead to unreliable extraction, as shown in Figure 2. The first situation occurs when a point (green) is extracted from the object edge line (orange), which is roughly parallel to the laser beam. This is treated as an unreliable laser return which will be excluded in the subregion. The second situation is to select the feature on the boundary of an occluded region. For example, Figure 2b shows the failure feature point (green) selected by its curvature. In this situation, it can be clearly identified that the selected point is not the edge point and is connected with other surfaces (dot line). This is because the laser beams are blocked by another object. Once the LiDAR moves to the left-hand side, the occluded part becomes observable and is not considered as the edge point again.

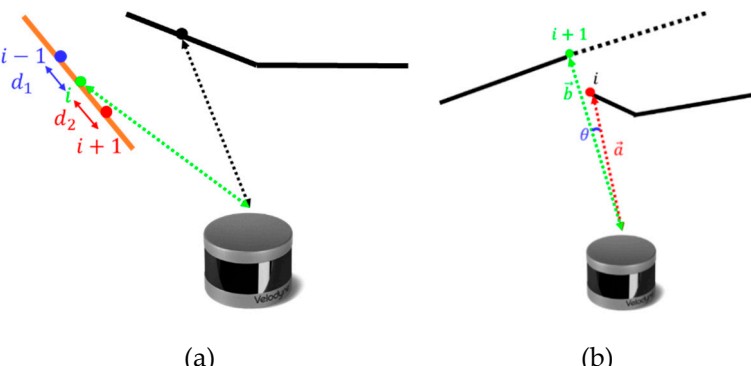

(a)                              (b)

**Figure 2.** Unreliable extraction from two situations: (**a**) roughly parallel situation, (**b**) occluded situation.

To prevent the occurrence of such a situation, the first strategy for solving the first situation is to calculate the distances between consecutive points ($d_1$, $d_2$). LiDAR's horizontal resolution is fixed. If the distances are higher than the threshold, it means the laser beam's direction is somewhat parallel with the object edge line or object surface. The second strategy is to prevent the occluded situation. By calculating the vectors and distances between points and LiDAR's center ($\|\vec{a}\|$, $\|\vec{b}\|$, $\vec{a}$, $\vec{b}$), if the offset of two distances is larger than the threshold and the angle between two vectors is smaller than the threshold, they are considered as unreliable features.

After the prevention of these two situations, the features extracted from the point can be evenly distributed and are more reliable for future LiDAR odometry use. Figure 3 illustrates the features extracted from the underground parking lot, where the red points (Figure 3a) represent the edge points and the yellow points in Figure 3b show the planar points. These edge features are located around the corner or the edge line of the objects, while the planar points are selected equally within the environment.

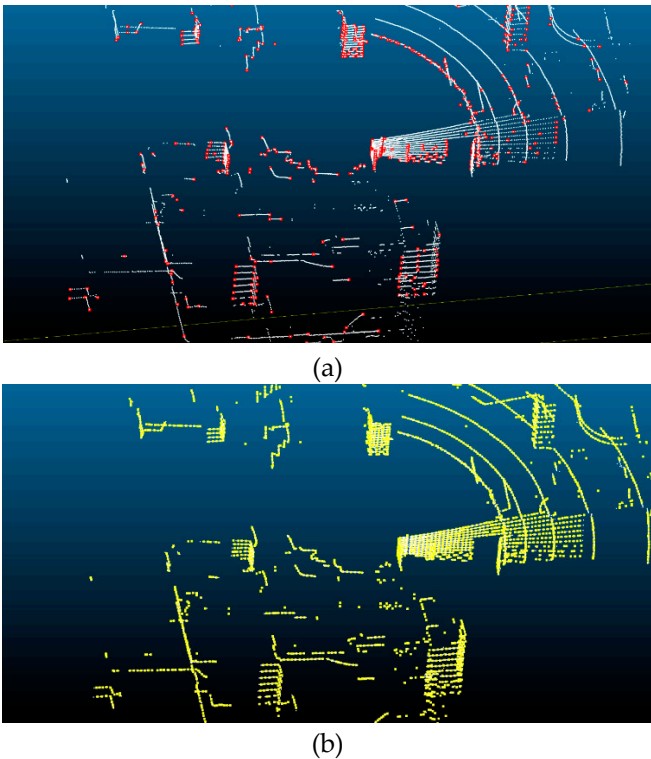

(a)

(b)

**Figure 3.** Demonstration of feature extraction: (**a**) edge points (red), (**b**) planar points (yellow).

2.2.2. LiDAR Odometry

LiDAR odometry plays the role of motion model in the proposed INS-aiding 3D LiDAR-SLAM. Odometry provides the relative position or velocity for the mapping process. For aligning each extracted feature to the previous feature, the first step is to find the corresponding feature in each scan. Based on the geometric relationship between each correspondence, the objective function can be built afterward. The relative position and attitude are estimated by accounting for the whole geometric relationship in the objective function. Therefore, LiDAR odometry can be separated into three parts: finding correspondence, formulating the objective function, and motion estimation [32].

First, it is important to define the types of geometric relationships that are used in LiDAR odometry. In this paper, two geometries are used: edge line, as the correspondence for an edge point, and planar surface, as the correspondence for a planar point. Figure 4 represents the geometric relationship defined in 3D LiDAR-SLAM. In Figure 4a, the edge line is composed of two edge points ($j_{k-1}$ and $l_{k-1}$) from the previous scan ($k-1$), which are the closest to the current edge point ($i_k$) and must be selected

in different channels. This is to prevent the selected edge line located on the same surface, which is parallel to the same scanning channels. The corresponding planar surface can be selected based on the three closest neighbors ($j_{k-1}$, $l_{k-1}$, and $h_{k-1}$) of the current planar feature ($i_k$). Figure 4b clearly shows that the planar surface (gray triangle) is determined by three correspondences and at least two planar points in different channels.

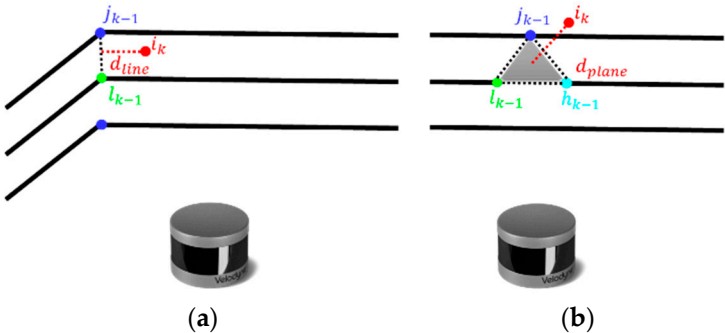

**Figure 4.** Geometric relationship for extracted features: (**a**) edge line for an edge point, (**b**) planar surface for a planar point.

Second, the objective functions are built based on these two geometries. The edge point ($i_k$) is assumed to be aligned with the determined edge line, which means that the distance between the edge line and edge point should be zero. The point-to-line distance is defined as:

$$d_{line} = \frac{\|(\widetilde{p}_{k,i} - p_{k-1,j}) \times (\widetilde{p}_{k,i} - p_{k-1,l})\|}{\|p_{k-1,j} - p_{k-1,l}\|},$$
(6)

where $d_{line}$ is the point-to-line distance, $p$ represents the coordinates of the point, and $\times$ indicates the cross product. This equation indicates that the area of the parallelogram is equal to the base multiplied by the height. In other words, $\|(\widetilde{p}_{k,i} - p_{k-1,j}) \times (\widetilde{p}_{k,i} - p_{k-1,l})\|$ is the area of the parallelogram, represented by the magnitude of the vector cross product of two adjacent sides; the base is $\|p_{k-1,j} - p_{k-1,l}\|$; and the targeted $d_{line}$ is the height that can be calculated from division by the area and the base.

Similar to the concept of point-to-line distance, the point-to-plane distance is defined as:

$$d_{plane} = \frac{\|(\widetilde{p}_{k,i} - p_{k-1,l}) \cdot (p_{k-1,j} - p_{k-1,l}) \times (p_{k-1,j} - p_{k-1,h})\|}{\|(p_{k-1,j} - p_{k-1,l}) \times (p_{k-1,j} - p_{k-1,h})\|},$$
(7)

where $d_{plane}$ is the point-to-plane distance and $\cdot$ means dot product. In this equation, the volume of the parallelepiped is obtained by the multiplication of the base area and the height. For example, $\|(\widetilde{p}_{k,i} - p_{k-1,l}) \cdot (p_{k-1,j} - p_{k-1,l}) \times (p_{k-1,j} - p_{k-1,h})\|$ is the volume of a parallelepiped obtained using a combination of a cross product and a dot product. The base area is $\|(p_{k-1,j} - p_{k-1,l}) \times (p_{k-1,j} - p_{k-1,h})\|$, which is similar to Equation (6), and the determined $d_{plane}$ is the height from the volume divided by the base area.

The final part of LiDAR odometry is to estimate the relative position and attitude. In this SLAM algorithm, the motion of the sensor is assumed to be constant angular and linear velocities. As a result, the rigid transformation is written as:

$$p_{k,i} = C_{k-1}^{k} \times \widetilde{p}_{k,i} + T_{k-1}^{k},$$
(8)

where $p_{k,i}$ and $\widetilde{p}_{k,i}$ are the i-th coordinates of the feature points after and before transformation, $C_{k-1}^{k}$ is the rotation matrix from time $k-1$ to $k$, and $T_{k-1}^{k}$ is the translation vector.

Furthermore, the combined function for Equations (6) and (7) can be rewritten as:

$$f\left(\widetilde{p}_{k,i}, C_{k-1}^{k}, T_{k-1}^{k}\right) = d. \tag{9}$$

To solve the nonlinear function ($f$), the Levenberg–Marquardt (LM) algorithm is used [38], which is considered as a combination of the Gauss–Newton algorithm and the method of gradient descent, especially for addressing nonlinear least-square problems. The objective function for Equation (9) is written as:

$$(C_{k-1}^{k}, T_{k-1}^{k}) \leftarrow \left(C'^{k}_{k-1}, T'^{k}_{k-1}\right) + \left[J'^{T}J' + \lambda' diag\left(J'^{T}J'\right)\right]^{-1} J'^{T}d', \tag{10}$$

where ' indicates the estimated value based on each iteration, $J$ is the Jacobian matrix of $f$ with respect to the determined rigid transformation parameters ($C_{k-1}^{k}$, $T_{k-1}^{k}$), and $\lambda$ is the damping factor, which can increase or decrease according to the improvement in the solution.

### 2.2.3. LiDAR Mapping

LiDAR odometry is a process for registering features in the current scan from the previous features. In addition, the transformation during each scan is assumed to be a constant and linear movement. However, this assumption is not always appropriate, especially in the application of speedy land vehicles or drones, which means that the movement is usually nonlinear. LiDAR mapping plays a primary role in correcting the distorted movement from LiDAR odometry, as well as generating a consistent global map for each LiDAR mapping registration.

The concept of LiDAR mapping is to align the feature points with the global map, which is also an incremental map based on each LiDAR mapping registration [31]. LiDAR mapping extracts the corresponding points from the global map and matches them with the features from LiDAR odometry. Afterward, these features are stored in the global map, which gradually expands the map to enhance the future loop closure solution. However, it is time-consuming to conduct this global registration process. In this structure, LiDAR mapping is processed at a lower frequency, 4 Hz, while LiDAR odometry is processed at 10 Hz. The low processing frequency of LiDAR mapping might lead to the problem of losing dynamic information, as the vehicle is at a high speed. The SLAM-PVA model is proposed to address this issue in the following section.

To explain further, Figure 5 illustrates the mapping process. In the beginning, the correspondences from the global map (black line, $\overline{p}_{k-1}$) need to be extracted and 10 m cubic areas are built to store these correspondences. If the current features (red line, $p_k$) generated from LiDAR odometry intersect with the cubes, these correspondences are extracted and stored in a 3D KD-tree [39]. In brief, this process benefits the efficiency of the mapping process, preventing searching of the whole global map and only targeting the intersection area instead.

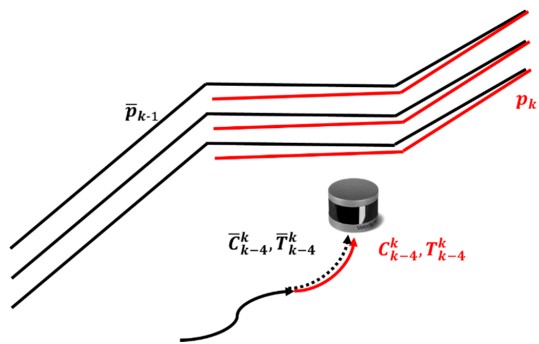

**Figure 5.** Demonstration of the LiDAR mapping process.

Next, the objective functions will be created based on these correspondences and current features according to the motion estimation in LiDAR odometry. The combined functions also include the edge

and planar features, which are solved using the LM method [38]. To prevent the least-square algorithm from sticking at the local minimum, the initial value from the LiDAR odometry is used ($C_{k-4}^k$, $T_{k-4}^k$) in the first iteration [31]. The final rigid transformation parameters ($\overline{C}_{k-4}^k$, $\overline{T}_{k-4}^k$) are corrected and estimated according to matching with the global map.

The final step of grid-based SLAM is the map updating when we have determined that the system needs it. Two thresholds are designed to trigger this step. The first threshold ($T_{Dist}$) is the distance travelled between the previous and current updates ($\Delta D$). If $\Delta D$ is larger than the $T_{Dist}$, the current grid map will be updated onto the existing global map or vice versa. Another condition is heading angle difference ($\Delta\psi_{az}$) with its threshold, $T_{Az}$. The updating process is to add the log odds of the current map to the existing map.

Generally, 3D LiDAR-SLAM runs properly in various situations. However, it is also limited by the speed and movement of the vehicle. In other words, as the LiDAR sampling rate is 10 Hz, if the speed of the vehicle is 70 km/h, the displacement between each scan (0.1 s) is about 2 m. In addition, it is normal for a land vehicle to accelerate and decelerate, particularly in an urban area. The major challenges of LiDAR-SLAM are an unreliable initial value and nonlinear movement. As a result, this paper proposes an INS-aiding structure for 3D LiDAR-SLAM, taking advantage of SLAM to improve the overall solution.

### 2.3. Integrated Navigation Structure for Automated Vehicles

As discussed in the previous section, LiDAR odometry and mapping can run properly by themselves without any external source. However, this is under conditions where the environment is clear, without many moving objects, and the LiDAR platform is moving slowly. As this paper focuses on speedy automated vehicles and trying to enhance navigation performance, it is necessary to expand the system's ability and make the overall LiDAR-SLAM more robust. Meanwhile, SLAM-derived measurements can also benefit the central fusion algorithm to provide more reliable and accurate navigation information. Figure 6 presents the detailed process of INS-aiding 3D LiDAR-SLAM (as shown in Figure 1, green box). As shown in Figure 6, this integration scheme first adopts fault detection exclusion (FDE) to eliminate the failure solution from LiDAR odometry through the INS solution. Similarly, FDE is also applied to validate the solution obtained from the LiDAR mapping process. If the final result from the mapping process is not stable or trustable, it will be excluded from the measurement update in SLAM-PVA EKF. Similarly, the refreshing process is utilized under this situation to refresh the map. Finally, SLAM-PVA EKF provides a smooth velocity instead of using a relatively position-derived velocity, as well as feeding the heading measurement from LiDAR mapping. In addition, this paper discusses the error model of the SLAM-derived measurement for the fusion filter. The details are provided as follows.

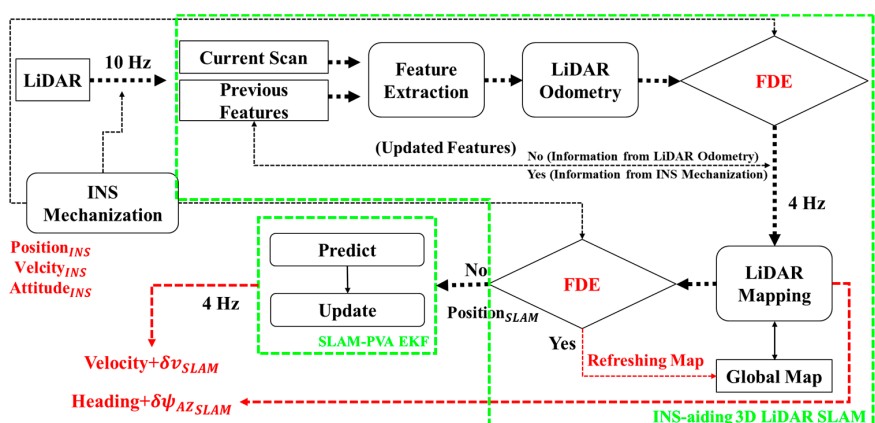

**Figure 6.** Structure of the proposed INS-aiding 3D LiDAR-SLAM and SLAM-position velocity acceleration (PVA) extended Kalman filter (EKF).

### 2.3.1. Fault Detection and Exclusion

FDE is an algorithm used in GNSS to improve the position solution quality [40]. FDE can check the accuracy and reliability of GNSS signals to detect outliers in GPS data and eliminate these outliers from the navigation solution. In the integration scheme, FDE is adopted to detect an unreasonable solution from LiDAR odometry and mapping. By continuously updating the navigation solution and feedback bias for INS, it is capable of monitoring other outlier measurements in a short period. According to this assumption, the fusion algorithm uses the INS velocity and integrated heading to detect whether the LiDAR odometry or mapping process works properly. In addition, it takes distances of the final corresponding points (edge and planar points) into account. This is another method for evaluating the performance of LiDAR-SLAM itself. Thus, there are three conditions to be evaluated in FDE.

First, the corresponding distance, including point-to-line and point-to-plane, should be smaller than the preset threshold ($\theta_{corDist}$) after the LM algorithm. $\theta_{corDist}$ is 0.10 meters, since the accuracy of the used LiDAR is 0.03 meters. The expected accuracy of SLAM follows the three-sigma rule of thumb. Specifically, the average residual error in Equation (9) should be less than a certain value. Second, the innovation sequence (innovation is the difference between the observed value and the forecast value) between the velocity derived from the relative position and the INS velocity must be less than $\theta_{velocity}$. The final condition is to monitor the heading information. If the heading innovation sequence is larger than $\theta_{heading}$, FDE would consider it as an unreliable solution. $\theta_{velocity}$ is 0.6 meters per second and $\theta_{heading}$ is 0.5 degrees, since these are the maximum tolerated values in the proposed integrated system.

$$\begin{cases} if \ \frac{\sum_i^w f\left(p_{k,i}, C'^k_{k-1}, T'^k_{k-1}\right)}{w} \leq \theta_{corDist} \\ \quad if \ Velocity_{inno} \leq \theta_{velocity} \\ \quad if \ Heading_{inno} \leq \theta_{heading} \end{cases} , \ is \ qualified$$

LiDAR odometry acts as a motion model for the SLAM algorithm and is based on dead reckoning (DR), which is the process of calculating the current position based on a previously determined position. Hence, if there is an outlier solution estimated from LiDAR odometry, the error will accumulate until the next mapping process. What makes matters worse is that LiDAR odometry will provide an unreliable initial solution to LiDAR mapping, which means that the mapping process also has great possibilities of suffering from the same problem.

Based on the data processing shown in Figure 6, once the result from LiDAR odometry is detected as an unreliable solution, there are two strategies to address. As LiDAR odometry is a continuous process to align the features from the current scan with the previous scan, if the LiDAR odometry performs unstably, the first strategy is to replace the LiDAR odometry solution with the INS solution. Using a more stable INS solution prevents the outlier from influencing the updating process, which means that the edge and planar points are correctly transformed and stored for the next aligning. The second strategy in the odometry process benefits LiDAR mapping, as INS provides the initial value ($C^k_{k-4}$, $T^k_{k-4}$) for the mapping process instead of using an unreliable odometry solution. Since the mapping process is at the frequency of 4Hz, the INS-derived rotation and translation should cover the period from $k$ to $k-4$. This strategy ensures that the mapping process has a relatively trustable initial value for conducting the LM algorithm [38].

In terms of FDE in the mapping process, it takes charge of evaluating the final solution from 3D LiDAR-SLAM. SLAM-PVA EKF uses the position from the mapping process to update the navigation states; if the mapping solution is not correct, this measurement will be excluded from the EKF update while the refreshing process is adopted to regenerate the new global map. In other words, all features in the global map are emptied after refreshing. The idea of the refreshing process will be discussed below.

### 2.3.2. SLAM-PVA EKF Model

The proposed SLAM algorithm is composed of two major matching processes to derive the relative position and attitude: odometry and mapping, which process data with a high and a low frequency,

respectively. LiDAR odometry deals with the data based on the LiDAR sampling rate, which is set at 10 Hz in this paper. On the other hand, LiDAR mapping is set at a lower frequency of 4 Hz to reduce the calculation consumption and increase the speed. Compared to LiDAR odometry, the frequency of the final solution is much lower, which leads to loss of dynamic information if the velocity is derived from the divided relative position by time interval.

Therefore, the fusion structure uses the dynamic model to estimate the velocity as the major measurement for central fusion EKF. The dynamic model for navigation states can be classified into two models, position-velocity (PV) and PVA. The PV model is expected for the situation in which the vehicle has a nearly constant velocity or changes velocity slowly [19].

However, the constant-velocity assumption is not appropriate for the ground vehicle, which needs to brake and accelerate all the time with varied accelerations, particularly in urban areas.

To sum up, the proposed fusion algorithm uses the PVA model to estimate the velocity with a stationary process (Gauss–Markov process) to describe the error behavior of acceleration [19]. The PVA model is written in one-dimensional discrete-time form as follows:

$$x_{k,SLAM-PVA} = \begin{bmatrix} r & v & a \end{bmatrix}^T_{9\times1}. \tag{11}$$

The transition matrix ($\Phi_{k-k+1,SLAM-PVA}$) is defined as:

$$\Phi_{k-k+1,SLAM-PVA} = \begin{bmatrix} 1 & \Delta t & \frac{1}{\tau^2}\left(e^{-\tau\Delta t} + \tau\Delta t - 1\right) \\ 0 & 1 & \frac{1}{\tau}\left(1 - e^{-\tau\Delta t}\right) \\ 0 & 0 & e^{-\tau\Delta t} \end{bmatrix}, \tag{12}$$

where $\tau$ is the inverse of the correlation time.

The continuous-time process noise covariance matrix is written as:

$$Q(t_k)_{SLAM-PVA} = \begin{bmatrix} 0 & 0 & 0 \\ 0 & 0 & 0 \\ 0 & 0 & 2\sigma_a^2\tau \end{bmatrix}, \tag{13}$$

where $2\sigma_a^2\tau$ is the spectral density and $\sigma_a^2$ is the noise variance of acceleration. The variance covariance matrix $Q_{k,PVA}$ with an approximate solution can be determined by a trapezoidal integration method from [14].

By continuously feeding the position measurements from LiDAR mapping, the SLAM-PAV model is able to correctly estimate the navigation information. The output rate of the SLAM-PVA velocity is 4 Hz, according to the frequency of the mapping process.

### 2.3.3. Error Model for 3D LiDAR SLAM Measurements

The error model is an important implication for recognizing the quality of the updating measurement. Two measurements from 3D LiDAR-SLAM are used in this paper. In terms of velocity, the final velocities are obtained from PVA EKF. Instead of using the covariance matrix from EKF, the proposed error model takes the geometry effects and residuals into account, as follows:

$$P^{SLAM} = \sigma^2 \times \left(J'^T J'\right)^{-1}, \tag{14}$$

$$\sigma^2 = \frac{\sum_i^w \left(f\left(p_{k,i}, C'^k_{k-1}, T'^k_{k-1}\right)\right)^2}{w}, \tag{15}$$

where $P^{SLAM}$ is the covariance matrix of six degrees of freedom (6DoF), including position and attitude, and $J'^T J'$ represents the geometry effects with respect to the dilution of precision (DOP) in GNSS.

In addition, $\sigma^2$ implies the residual of the corresponding features, calculating the sum of the residuals and dividing by the number of corresponding features ($w$).

The variance of the residuals represents the degree of convergence between the current and corresponding features. A higher variance means a lower precision of the SLAM solution. Moreover, the geometry matrix ($J'^T J'$) indicates the variance of each component that can be extracted from the diagonal of matrix. Consequently, the covariance matrices of velocity and heading are derived from Equations (14) and (15).

### 2.3.4. Refreshing Map

As mentioned in the previous section, SLAM is based on the relative positioning approach. Once the incremental global map goes adrift, the SLAM-derived solution will also be influenced by the drifted map after the registration. Hence, this paper proposes a strategy, called refreshing map, to address this situation [41].

Once the failure of LiDAR-SLAM is detected from FDE, the refreshing process will empty the features of the global map while adding new features based on the INS solution. Because the INS solution is continuously updated and corrected by GNSS, motion constraints, and SLAM-derived measurements, the application is more trustable compared to the single-SLAM solution. After refreshing the global map in SLAM, the accumulating error in each feature is reset, which means that it is more reliable for the next registration.

### 2.3.5. Integrity Assessment

This paper proposes an integrity assessment to validate that each measurement is qualified ahead of the central EKF update. As the SLAM-derived measurements are validated from FDE, the integrity assessment focuses on the GNSS measurement condition to remove the failure or error-contaminated solution and reinforces the detection of motion constraint based on the physical behavior of the vehicle.

- GNSS Measurement Condition

This condition relies on the innovation sequence in the update process, applying both position and velocity innovations. To avoid the failure of the GNSS update, both innovations should meet the requirement of this condition ($\leq \theta_{Position}$ and $\leq \theta_{velocity}$). At the same time, if SLAM-derived measurements are available, the offset between the SLAM-derived velocity and the GNSS velocity ($\Delta Velocity$) must be smaller than the velocity threshold.

$$\begin{cases} \quad if \ Position_{inno} \leq \theta_{Position} \\ \quad if \ Velocity_{inno} \leq \theta_{velocity} \\ if (SLAM) \ \& \ \Delta Velocity \leq \theta_{velocity} \end{cases} , \ is \ qualified.$$

- ZUPT/ZIHR Condition

To accurately detect the ZUPT/ZIHR condition, it is not sufficient to only use IMU raw data or velocity information. The SLAM-derived velocity is qualified and the INS velocity is corrected in each update process. Two conditions are involved in implementing ZUPT/ZIHR detection. If the SLAM-derived velocity is qualified and less than the pre-set threshold ($\theta_{ZUPT}$), it is recognized as the stationary state. The SLAM-derived velocity is more relatively reliable, as SLAM has the advantage of relative position calculation. If there is no SLAM information, the INS velocity is still used because of the continuous correction from each measurement.

$$\begin{cases} if (SLAM) \ \& \ Velocity_{SLAM} \leq \theta_{ZUPT} \\ \quad if \ Velocity_{INS} \leq \theta_{ZUPT} \end{cases} , \ ZUPT/ZIHR.$$

In brief, contemporary fusion designs involve different sensor measurements. However, it is necessary to have a robust evaluation to prevent incorrect measurement in the update process. Based

on the concept of integrity, this paper proposes an integrity assessment to detect each measurement, even pseudo measurement, ahead of the update process. By applying this assessment, the central fusion algorithm can make sure that only good measurements are used. Even if there is no update or reliable sensor measurement in a short period, the fusion algorithm still provides a stable solution because of the well-designed INS model.

## 3. Field Testing

This paper presents the results from two major testing fields—GNSS-hostile regions with severe multipath interference or NLOS reception, and highway areas with long-term GNSS outages and high-speed movement. The primary purpose of this paper is to evaluate the overall 3D navigation performance in varied scenarios, especially focusing on height performance.

### 3.1. Configuration Description

In the designed vehicle, there are two IMUs, iNAV-RQH and PwrPak7D-E1 (Epson G320N), as ground truth and a testing system, respectively. PwrPak7D-E1 is a commercial device from NovAtel, including a tactical-grade IMU (Epson G320N) and GNSS receiver. RQH is a navigation-grade IMU which can achieve a 30 cm accuracy combined with a differential GNSS and less than 0.1% error in distance travelled with odometer. Moreover, RQH can maintain a 5 mm/s accuracy in the velocity domain and a 0.01 degrees accuracy in heading with a differential GNSS [42]. In this paper, RQH is combined with the differential GNSS from PwrPak7D-E1 and velocity information from on-board diagnostics-II (OBDII) integrating through commercial software (Inertial Explore). The reference information is processed using the tightly coupled (TC) scheme with a forward-backward smoother. Therefore, it is a trustable reference system based on the specification of manufacturing. On the other hand, the testing system uses two integration methods with real-time kinematic (RTK), which is a real-time GNSS solution obtained from PwrPak7D-E1. To simulate the real-time solution, it is adopted in the integrated design instead of using a post-process differential GNSS solution. The first integrated design is a conventional design, INS/RTK/Odometer (PwrPak7D-E1/OBDII), and the other one is the proposed method, INS/RTK/LiDAR-SLAM (PwrPak7D-E1/VLP-16). The configuration of the designed vehicle is shown in Figure 7 and the specification of IMU is given in Table 1. The LiDAR used in this paper is VLP-16, which is deployed on the top of the vehicle (Figure 7a). For comparison, all the solutions are synchronized with GPS time and the relationships of each sensor are measured by the surveying tapes (Figure 7b,c). The following statistical tables were generated using this information to evaluate the performance of the proposed algorithm.

**Table 1.** Specifications of the iNAV-RQH (reference system) and Epson G320N.

|  | iNAV-RQH | |
| --- | --- | --- |
|  | Accelerometer | Gyroscope |
| Bias Instability | <15 $\mu g$ | <0.002°$/hr$ |
| Random Walk Noise | 8 $\mu g/\sqrt{Hz}$ | 0.0018°$/\sqrt{hr}$ |
|  | PwrPak7D-E1 (Epson G320N) | |
|  | Accelerometer | Gyroscope |
| Bias Instability | ≤100 ug | 3.5°$/hr$ |
| Random Walk Noise | ≤0.5 $m/s/\sqrt{hr}$ | 0.1°$/\sqrt{hr}$ |

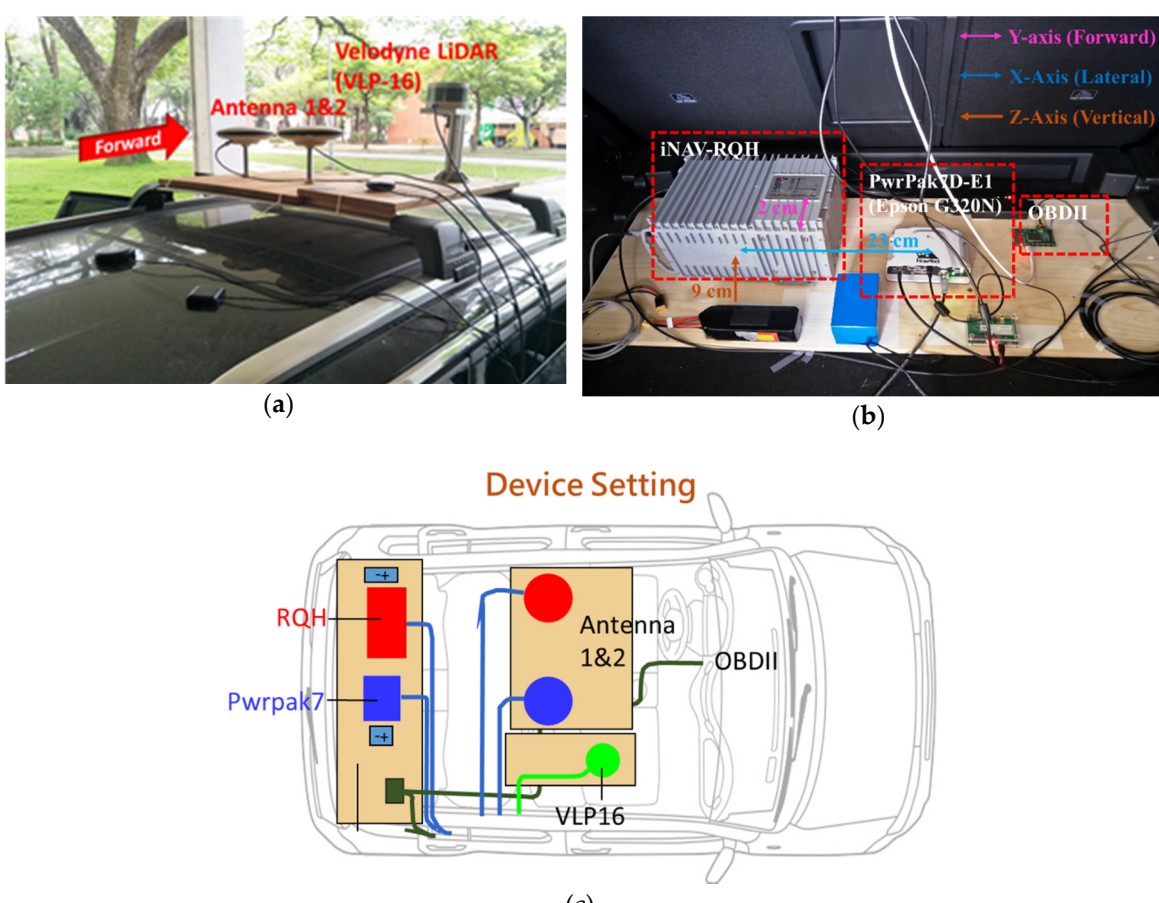

**Figure 7.** Configuration of the designed land vehicle: (**a**) GNSS antennas and LiDAR, (**b**) inertial measurement units (IMUs), (**c**) device setting.

*3.2. Scenario Description*

Two scenarios are presented in this paper. The first one is a GNSS-hostile area, which includes the GNSS-denied environment and several experimental routes surrounded by high buildings. The second scenario is a highway area, which involves various height changes and long-term GNSS outage.

3.2.1. Scenario 1: GNSS-Hostile Region

Scenario 1 is selected in a highly urbanized area with several surrounding high buildings. The experiment is conducted in the open sky area to have a good initial alignment of the reference system. In this experiment, GNSS suffers from severe signal blockage and multipath interference or NLOS reception. Figure 8 shows the number of satellites from the differential GNSS solution. The color of each solution is encoded by the number of satellites, which indicates how challenging the testing field is. The block frames demonstrate the worse parts of the testing field, where it is highly influenced by the multipath interference and NLOS reception. The following section will discuss these areas.

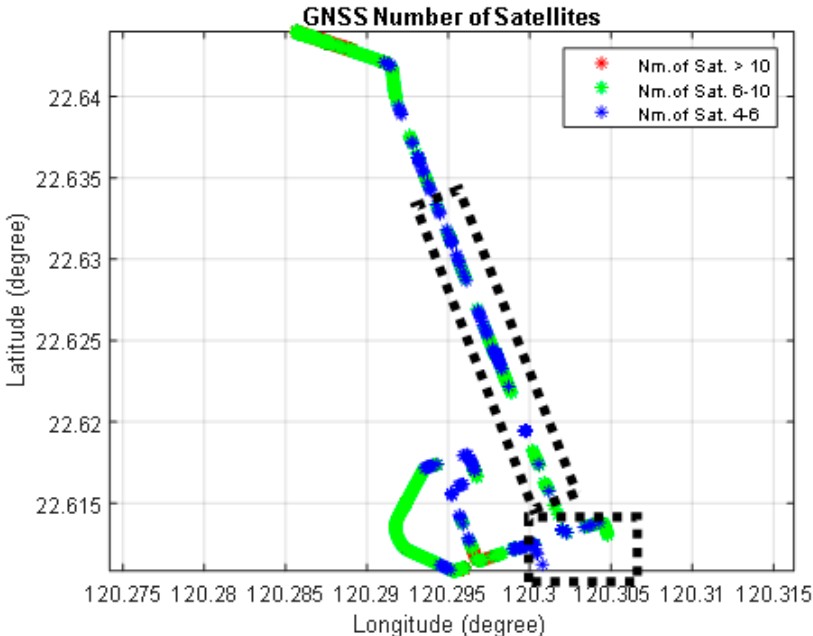

**Figure 8.** Differential GNSS solution encoded by the number of satellites in an urban area.

3.2.2. Scenario 2: Highway Area

The second scenario is conducted along with the highway. There are several up-and-down paths to test the height performance and the long-term under-highway paths. Furthermore, the speed is maximally up to 100 km/h during driving on the highway, which tests the robustness and flexibility of the proposed integrated method. Figure 9 represents the overall differential GNSS solution with information from the number of satellites. Two block frames indicate the areas where the vehicle drives under the highway. The outage is up to 345 s with a 1.5 km traveled distance and 477 s with a 4.7 km traveled distance at a very high speed.

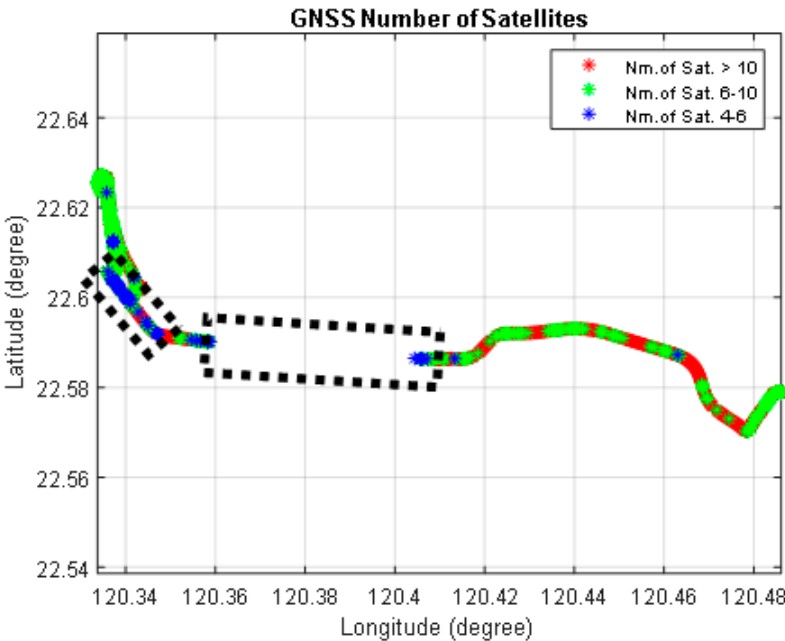

**Figure 9.** Differential GNSS solution encoded by the number of satellites in a highway area.

## 4. Results and Discussion

The results presented in this paper focus on two scenarios, GNSS-hostile regions and highway areas. In the first scenario, the LiDAR-SLAM-integrated design exhibits a considerable improvement compared to the conventional INS/RTK/odometer. Even in a deep urban area, the result presents a stable and reliable navigation performance, as well as the ability to overcome multipath interference or NLOS reception.

In the second scenario, the result along with the highway shows that the integrated system can correctly estimate the height information irrespective of the vehicle being underneath or on the highway. INS/GNSS/3D LiDAR-SLAM can deal with different dynamic movements, such as high- or low-speed vehicles.

### 4.1. Scenario 1: GNSS-Hostile Region

As discussed in the scenario description, the overall GNSS solutions in this experiment are mostly influenced by the blocked and error-contaminated signal, which leads to several outages and incorrect GNSS solutions. Tables 2 and 3 provide a statistical analysis. As seen from the root mean square error (RMSE), INS/RTK/3D LiDAR-SLAM is able to achieve a positioning accuracy of less than 1 m in three dimensions. Compared to the conventional INS/GNSS/odometer integration [11], the proposed method shows considerable improvement: 73% in north, 71% in east, and 77% in height. The proposed integrity assessment contributes the positioning improvement in scenario 1 which protects the update process from using error-contaminated GNSS. On the other hand, the velocities are also enhanced with 52% in north, 17% in east, and 2% in vertical direction. Although the improvement in vertical velocity is only 2%, the overall navigation performance in height is remarkably enhanced. The error-contaminated GNSS (especially in height domain) is effectively removed. The following description provides a detailed analysis based on the GNSS-hostile region.

**Table 2.** Statistical analysis of the INS/ real-time kinematic (RTK)/odometer.

| | | | INS/RTK/Odometer | | | | |
|---|---|---|---|---|---|---|---|
| Error | North (meter) | East (meter) | Height (meter) | North Velocity (m/s) | East Velocity (m/s) | Vertical Velocity (m/s) | Heading (degree) |
| Mean | −1.072 | −1.276 | 1.698 | −0.011 | 0.001 | 0.001 | 0.281 |
| STD | 2.268 | 2.739 | 2.246 | 0.057 | 0.052 | 0.120 | 0.151 |
| RMSE | 2.509 | 3.021 | 2.815 | 0.059 | 0.052 | 0.120 | 0.314 |
| Max. | 9.747 | 8.389 | 7.531 | 0.348 | 0.277 | 1.392 | 0.567 |

**Table 3.** Statistical analysis of INS/RTK/3D LiDAR-SLAM.

| | | | INS/RTK/3D LiDAR-SLAM | | | | |
|---|---|---|---|---|---|---|---|
| Error | North (meter) | East (meter) | Height (meter) | North Velocity (m/s) | East Velocity (m/s) | Vertical Velocity (m/s) | Heading (degree) |
| Mean | −0.088 | −0.252 | 0.198 | 0.001 | 0.001 | −0.002 | 0.256 |
| STD | 0.678 | 0.836 | 0.591 | 0.028 | 0.043 | 0.118 | 0.121 |
| RMSE | 0.684 | 0.873 | 0.623 | 0.028 | 0.043 | 0.118 | 0.283 |
| Max. | 3.184 | 3.507 | 2.906 | 0.223 | 0.337 | −0.002 | 0.564 |
| Improvement | 73% | 71% | 77% | 52% | 17% | 2% | 10% |

As shown in Figure 10, there are two figures—horizontal trajectory and height profile. The reference is plotted by the red line, the green line represents the integration result with the INS-aiding LiDAR-SLAM, the blue line indicates the trajectory from the conventional INS/GNSS/odometer, and the orange point shows the GNSS solutions. According to Figure 10a, the maximum errors in both

north and east directions occur in the multipath interference or NLOS reception area, which are up to approximately 3.50 (black circle) and 3.18 (cyan circle) m, respectively. As seen from the height profile, the general height solution from GNSS is untestable in the GNSS-hostile area. However, we can clearly see improvement from Figure 10b. The height of the proposed method is much closer to the reference, with around 0.6 m in RMSE and 63% improvement, while the conventional positioning method is heavily influenced by the error-contaminated GNSS solutions, with around 10 m maximum error.

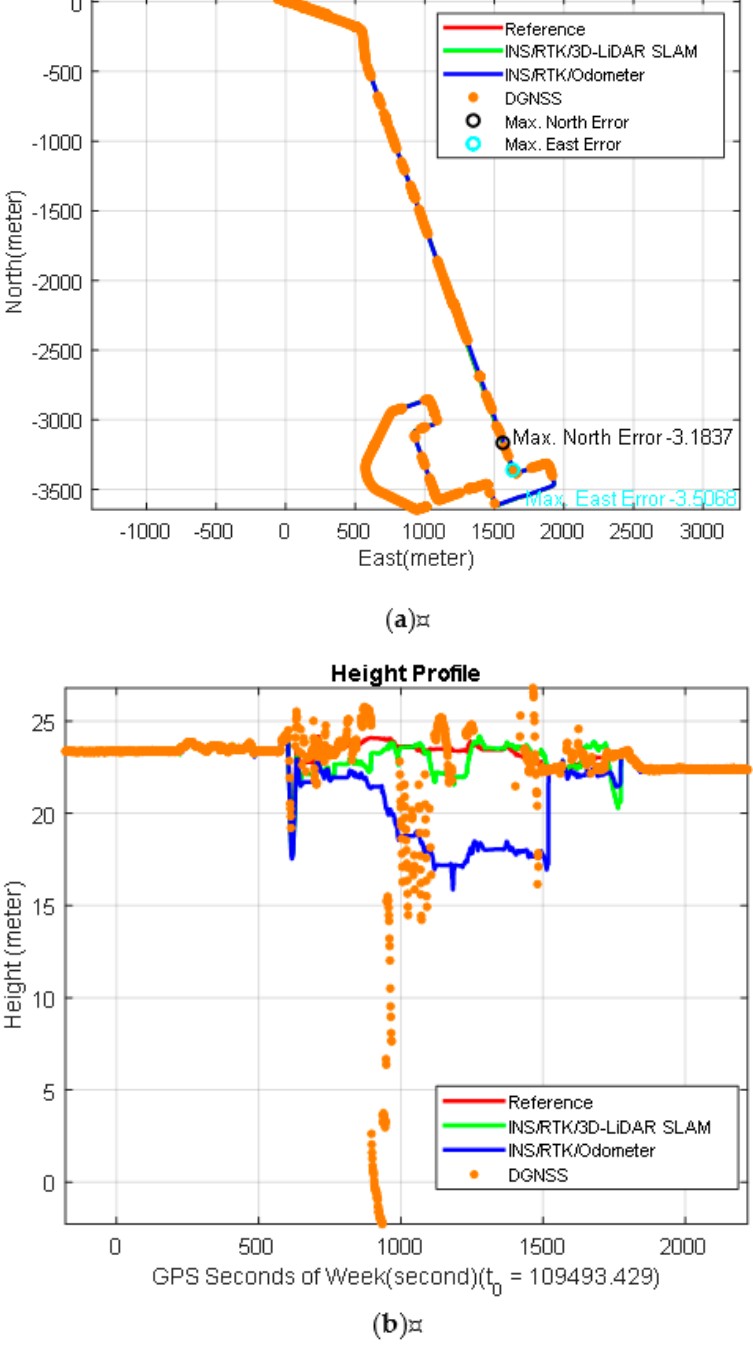

**Figure 10.** Scenario 1 results: (**a**) horizontal trajectory, (**b**) height profile.

Figures 11 and 12 show the zoomed GNSS-hostile and GNSS-denied in scenario 1 to concentrate on the performance of the proposed method. In Figure 11, the red blocks emphasize the error-contaminated GNSS. Clearly, the orange points deviate from the reference, even located 5–10 m away. In this situation,

the conventional positioning method cannot avoid the wrong GNSS solution, as the accuracy index from the GNSS system is not correct. In other words, the GNSS system does not detect the impact of multipath interference or NLOS reception and still gives an unreliable solution. In contrast, the integrated design takes advantage of each measurement and conducts an integrity assessment to remove failure measurements, especially from GNSS. As a result, the possibility of taking failure measurement in the update process significantly reduces. Moreover, the stable velocity and heading measurements from the INS-aiding SLAM benefit the integrated system. In general, the integrated approach sticks closer to the reference, even when there are several GNSS outages, as shown in Figure 12. The INS/GNSS/odometer maintains a consistent performance. However, the accuracy degrades with the travel distance due to the error behavior of the odometer.

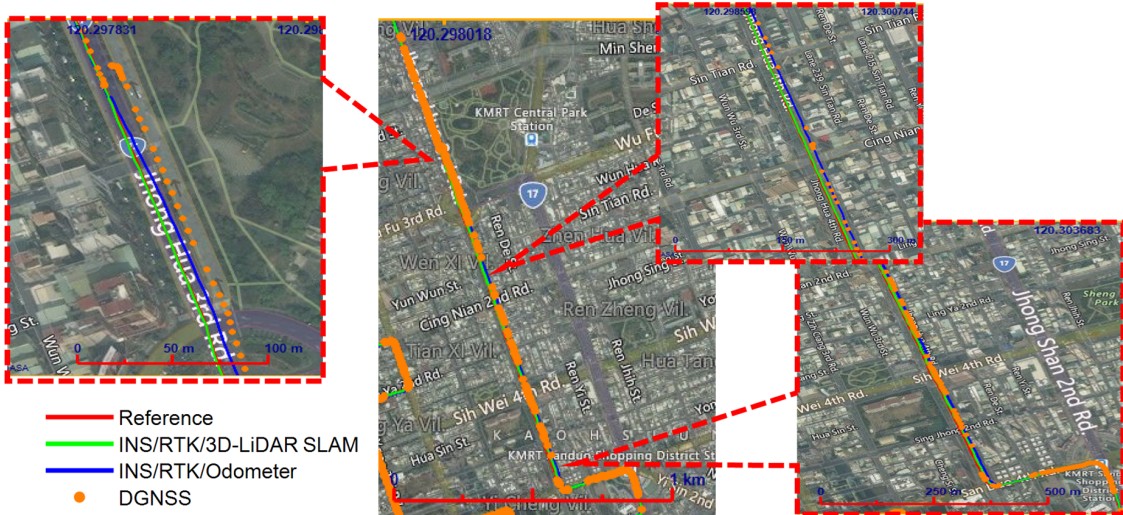

**Figure 11.** Scenario 1: zoomed GNSS-hostile area with severe multipath interference or non-line-of-sight (NLOS) reception.

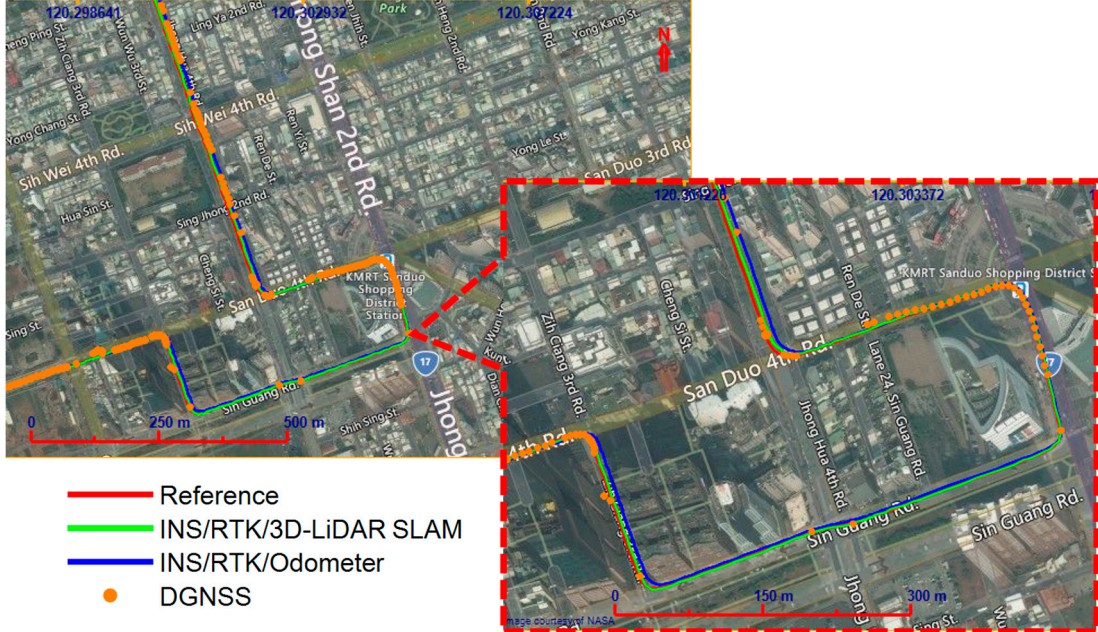

**Figure 12.** Scenario 1: zoomed GNSS-denied area.

## 4.2. Scenario 2: Highway Area

Scenario 2 is much more challenging than scenario 1, and involves long-term outages and impacts on the multipath while the vehicle is underneath the highway. In addition, the high-speed movement leads to the divergence of SLAM solutions. Table 4 gives the performance of conventional INS/RTK/odometer results. Thx maximum error is in the east direction (around 21 meters), which occurs during the long-term GNSS-denied area. In summary, the performance of the proposed method still achieves a 1 m accuracy, although the RMSE in east is up to approximately 1.2 m. However, a remarkable improvement is shown in Table 5. In such a SLAM-unfriendly situation, there are 39%, 77%, 47%, and 7% improvements in the north, east, height, and heading, respectively. Moreover, the velocities are also enhanced, which achieves more than 20% improvement in the horizontal direction and around 9% in the vertical direction.

**Table 4.** Statistical analysis of INS/RTK/odometer.

| | INS/RTK/Odometer | | | | | | |
|---|---|---|---|---|---|---|---|
| Error | North (meter) | East (meter) | Height (meter) | North Velocity (m/s) | East Velocity (m/s) | Vertical Velocity (m/s) | Heading (degree) |
| Mean | 0.105 | 2.387 | 1.083 | 0.005 | 0.005 | 0.001 | 0.167 |
| STD | 1.505 | 5.192 | 1.315 | 0.038 | 0.060 | 0.235 | 0.294 |
| RMSE | 1.509 | 5.714 | 1.703 | 0.038 | 0.060 | 0.235 | 0.338 |
| Max. | 6.780 | 21.426 | 4.638 | 0.272 | 0.341 | 2.179 | 0.617 |

**Table 5.** Statistical analysis of INS/RTK/3D LiDAR-SLAM.

| | INS/RTK/3D LiDAR-SLAM | | | | | | |
|---|---|---|---|---|---|---|---|
| Error | North (meter) | East (meter) | Height (meter) | North Velocity (m/s) | East Velocity (m/s) | Vertical Velocity (m/s) | Heading (degree) |
| Mean | 0.146 | 0.328 | 0.537 | 0.001 | 0.001 | −0.001 | 0.152 |
| STD | 0.916 | 1.247 | 0.716 | 0.028 | 0.043 | 0.214 | 0.276 |
| RMSE | 0.927 | 1.289 | 0.895 | 0.028 | 0.043 | 0.214 | 0.315 |
| Max. | 5.597 | 6.999 | 3.626 | 0.223 | 0.337 | 2.164 | 0.538 |
| Improvement | 39% | 77% | 47% | 26% | 28% | 9% | 7% |

Generally, there are two long-term GNSS outages in scenario 2. The maximum error occurs during the first outage, which is indicated in Figure 13a. As the vehicle directly goes to the area under the bridge (first outage), the EKF does not have enough observability to correctly estimate the bias and scale of the gyroscopes and accelerometer. Although the integrated system has aid from SLAM, it still causes the maximum error in the situation. Figure 13b illustrates the height profile, which indicates the outage duration in the black frame and height changes in the cyan frame. This figure clearly indicates that the height fluctuates because the vehicle continuously changes the path to move up and down along with or on the highway. The proposed integrated design performs well, especially in terms of height. Except for the first outage, the green line in the height profile is much closer to the reference, while the blue line has an approximately 2 to 3 m error during the same outage.

Figure 14 shows a zoomed picture of the first outage, which lasts for over 5 min with a distance of 1.5 km. By using the reference to evaluate, we can see clearly that the INS/GNSS/odometer result has a severe drift, while the proposed method has a better performance. As the integrated system adopts the SLAM-derived heading information, even in such a long-term outage, the overall 3D navigation performance is maintained within 4 m. The first outage loop closure error in percentage (drift error with traveled distance) is approximately 0.3%. In other words, the integrated system can provide a 3 m accuracy in 3D with a 1 km traveled distance; the central EKF in the first outage is not stable to

converge. As shown in Figure 15, the second outage is longer irrespective of the time or distance, which means that the vehicle moves more quickly than that in the first outage. Under this condition, pure SLAM might suffer from an incorrect initial value. However, the INS-aiding method contributes to SLAM. Even if the solution from SLAM diverges, the FDE process can detect the failure solution and use the refreshing process to reset the error. Moreover, EKF is converged by conceiving a reliable GNSS solution before going into the second outage, as the vehicle travels on the highway for a while. The loop closure error in percentage in the second outage is much better than that in the first one—only 0.1%. This means that even with the long-term outage and speedy movement, the integrated design allows the user to have a 1 m accuracy with a 1 km traveled distance.

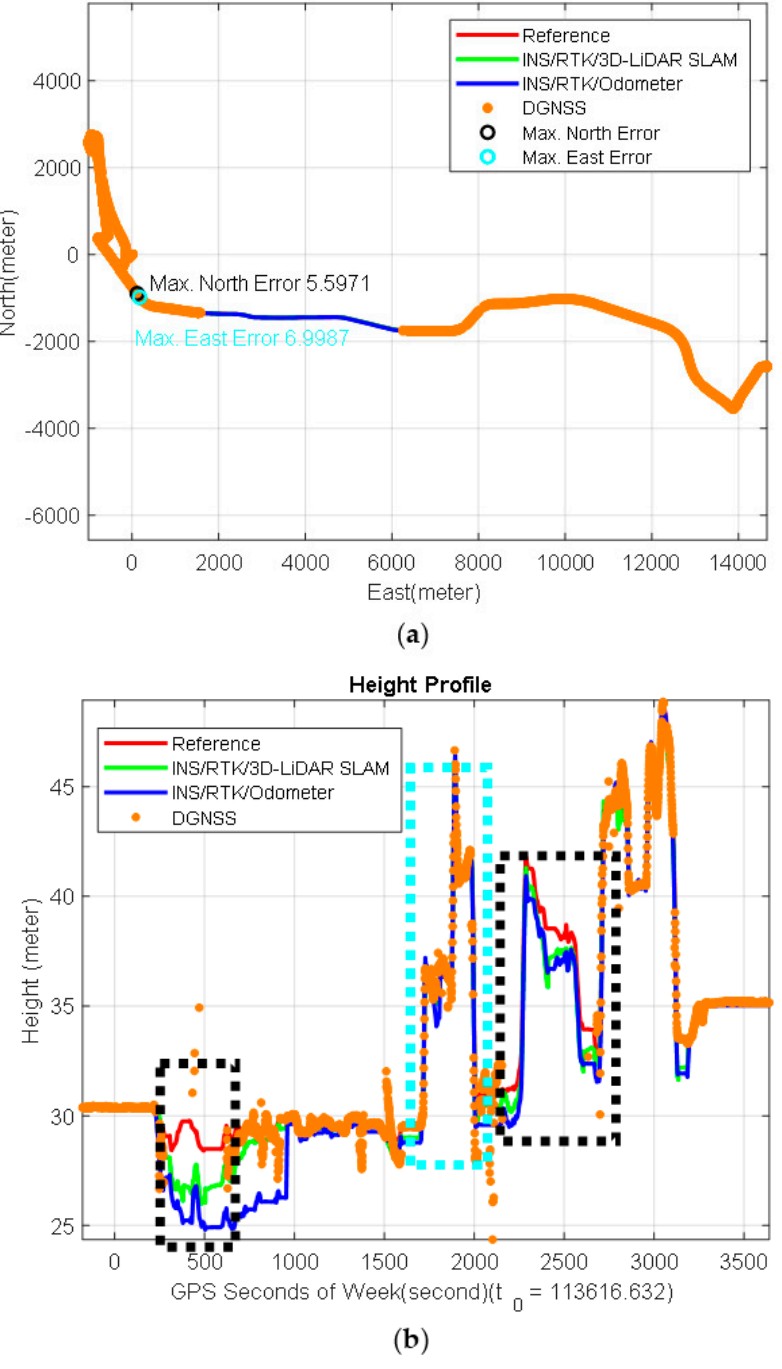

**Figure 13.** Scenario 2 results: (**a**) horizontal trajectory, (**b**) height profile.

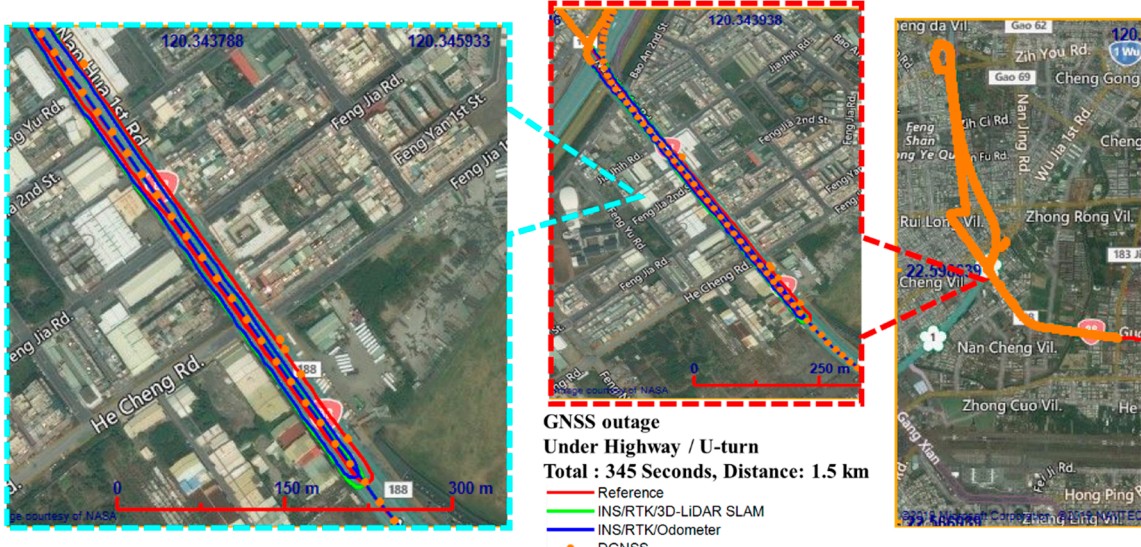

**Figure 14.** Scenario 2: zoomed area underneath the highway with a long-term GNSS outage and U-turn.

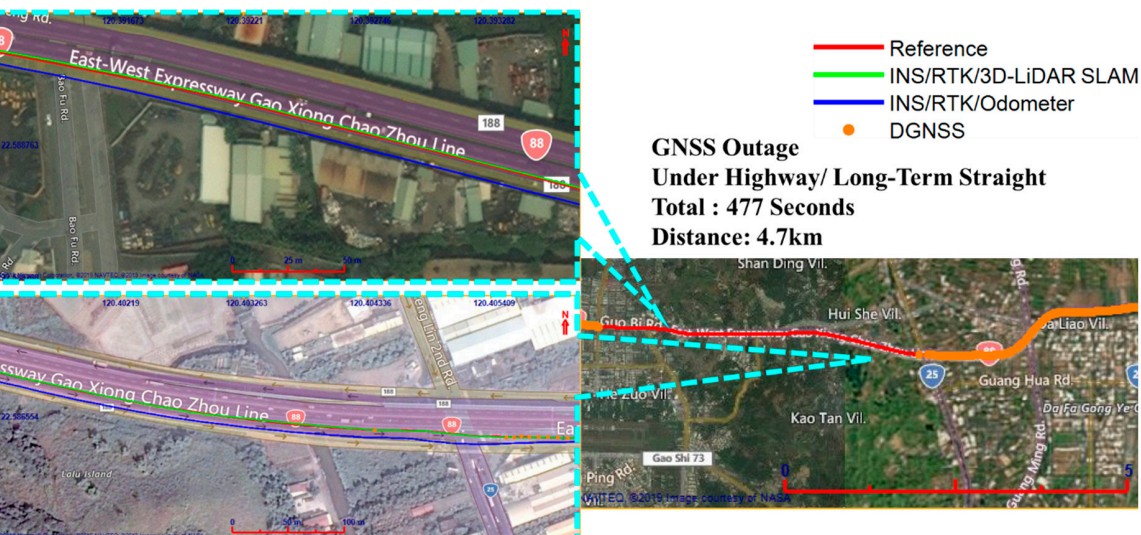

**Figure 15.** Scenario 2: zoomed area underneath the highway with long-term straight movement without a GNSS solution.

## 5. Conclusions

This study investigated an integrated design combining INS/GNSS/3D and LiDAR-SLAM in varied environments using a real-world dataset. There are two significant improvements for the integrated design compared to the conventional positioning method (INS/GNSS/odometer).

The first contribution is the proposed INS-aiding SLAM to overcome the drawbacks of pure SLAM. FDE is used to increase the stability of SLAM while removing the failure solution. The integration process uses SLAM-PVA measurements with a reliable error model in the central fusion filter. The second contribution is the integrity assessment, which allows the system to be more robust by avoiding failure measurements from each sensor. This is particularly relevant for the GNSS solution contaminated by multipath interference or NLOS reception, since the accuracy index from the GNSS system is not correct.

In general, the presented INS/GNSS/3D LiDAR-SLAM method can achieve a 1 m 3D accuracy in various environments or situations based on the results from GNSS-hostile and GNSS-denied areas.

Compared to the conventional positioning method, the results show an over 60% improvement in horizontal performance and around a 50% improvement in height.

In terms of navigation system, there are some strategies that can improve the performance. The proposed algorithm only involves a loosely coupled (LC) scheme. It is worth developing the TC scheme of multi-sensor fusion, especially in situations where is hard to access more than four satellites. In addition to using different fusion schemes, involving more sensors is also an alternative for enhancing the positioning performance, such as using barometers to control the height drift, magnetometers to mitigate the heading drift, and wireless sensors to provide position updates in indoor environments. For evaluation, the individual integration methods also need to be evaluated separately and presented thoroughly to indicate the main contributions.

**Author Contributions:** Conceptualization, K.-W.C. and G.-J.T.; methodology, G.-J.T. and Y.L.; validation, Y.-H.L.; formal analysis, G.-J.T. and Y.L.; data curation, Y.-H.L.; writing—original draft preparation, G.-J.T.; writing—review and editing, K.-W.C. and N.E.-S.; visualization, G.-J.T.; supervision, K.-W.C. and N.E.-S. All authors have read and agreed to the published version of the manuscript.

**Funding:** This research was funded by Ministry of Science and Technology, grant number 107-2221-E-006-125-MY3 and 108-2917-I-006-005.

**Acknowledgments:** The authors thank Ministry of Science and Technology (MOST) for its financial support and the project under the Ministry of Interior (MOI). They also thank the editor and anonymous reviewers for their constructive comments on this paper.

**Conflicts of Interest:** The authors declare no conflict of interest.

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
