# Peer review of "Navigation Engine Design for Automated Driving Using INS/GNSS/3D LiDAR-SLAM and Integrity Assessment"

_remotesensing, doi:10.3390/rs12101564_

Round 1
Reviewer 1 Report
Dear Authors,
I have read your paper entitled: Navigation Engine Design for Automated Driving Using INS/GNSS/3D LiDAR-SLAM and Integrity Assessment.
The paper is really interesting, actual and well written. I have no serious comments, but only a few questions, I would like to know the answers.
1) How was the INS aligned to the direction of movement? It means how was the sensor frame aligned with the vehicle frame?
2) Have you considered the lever arm between the GNSS antenna and INS?
3) How was the unit mounted inside the vehicle and how was it protected to the movement of the platform with INS related to the vehicle?
4) In Figs. 10, 13 you have shown the trajectory, but the errors should be also added as separated figures to see the error propagation of all information sources and the fusion.
At the and, I would like to write, that the current state of the art is well written, the 20 citations are newer than 5 years and the result section is very interesting and based on real data.
Best regards,
Reviewer
Reviewer 2 Report
The paper presents an approach for positioning and navigation of automated driving using INS/GNSS/3D LiDAR-SLAM. It is possible to perceive an interesting contribution to the area. However, in a general evaluation, the paper has one main weak spot: the experiments have some problems with their methods.
About the experiments, it was implemented two different systems: a reference and one with the approach proposed at this work. The problem is that each system was implemented in different hardware, which can lead to a miss evaluation of the generated results. Besides that, it was not presented any information about the software aspects of the reference system such as its internal architecture. Section 4 describes the results talking about improvements, however, it is difficult to evaluate these improvements since it was not clearly defined a reference system and there are pieces of evidence that we are dealing with different systems. In Conclusion, the authors comment about a "conventional positioning method", however, it was never defined in the whole text.
Still talking about a general evaluation, some important concepts to understand the paper content must be presented:
- registration process
- objective function
- edge point
- planar point
- surface point
- dead reckoning
- innovation sequence
- pseudo-measurements
- ZUPT/ ZIHQ condition
Finally, about the general comments, I suggest a more detailed description that would permit us to understand the differences between figures 1 and 6. Both figures describe the paper contribution, but they are different. Why are necessary two figures representing the same work? How are they complement each other?
After these general comments, I have some questions about specific issues for each section.
The Abstract did not any mention the problem approached in this work, to know: how to mitigate GNSS outage and interference in multisensor fusion and how to minimize error in a LIDAR arrangement in environments with moving objects in a vehicle under high speeds? The contributions of the paper are presented in general terms such as "improve the performance and increase the robustness to adjust to varied environments", and "(the method) contributes SLAM to eliminating the failure solution" without any words about what "failure solutions" means in this context. Also, the authors make the case that the "fusion structure enhances the conventional positioning algorithms by compensating for individual drawbacks" without any context about these drawbacks. At the end of the abstract, it would be interesting some words evaluating the contribution of the works for the research field.
The Introduction starts with a paragraph without any clear relationship with the paper content. It discusses V2V and IoT. I suggest excluding it.
In line 54, it is written "... The accuracy of these measurements is highly dependent on the environment." It is necessary a reference for this kind of assertion.
Between lines 109 and 110 there is a line break that I believe is a mistake. Apparently, both lines pertain to the same paragraph.
Also, in the same paragraph, when describing the paper contributions and is presented the Integrity Assessment, just a few words are used. Since it is one of the main contributions of the paper, a more detailed description of this method should be made.
At lines 137 and 144, it is said that a linearization process was carried out. I suggest a few words about the conditions of this linearization. In which conditions it can be applied.
In line 154, What it means NHC? In fact, I think that MDPI templates demand a list of acronyms that were not built.
A figure could aid to equation 5 understanding.
The terms described in line 252 are not in the previous equation (9).
I believe that the correct form for line 264 would be: "...features in the current scan from the previous features."
In line 287, when citing the LM method, the reference [39] must be indicated.
The equation presented in subsection 2.3.1 (pg 10) can be replaced by the whole algorithm described there.
In line 398, there is a "the" extra in the middle of the phrase.
I suggest a review in the text of the item 2.3.4. All features are cleaned in each refresh process? A whole new SLAM is stored? I consider an important issue since this module is one of the main contributions of the work.
In section 3, subsection 3.2.1, how the signal blockage and multipath interference was detected? How to measure these conditions?
The improvements indicated in Table 2 and 3 was calculated based on previous works. I suggest including those measures in the table for a better explanation of the results, and not just cited the reference.
About all the figures with the tests, in none of them is possible to visualize the reference line. Also, there are two Figure 13(a).
Finally, a more detailed evaluation of the performance of the modules that are the main contributions of the work should be presented, to know, FDE and integrity assessment.
Finally, I suggest a strong review of the bibliography, once there are some references without any source, like [16].
Reviewer 3 Report
The manuscript has been well written. However, the following should be reflected in the manuscript.
- Most of the parts of the proposed comes from the references. The authors does not describes details since they are in the references. Please clearly describe what the authors have proposed. The proposed parts should be described in more detail.
- In the introduction, the authors described that the targets for the manuscript are multipath interference or NLOS reception. Please describe that the proposed algorithm reduces the effect of the multipath interference or NLOS reception in a quantitative manner.
- Equation (9) and description of line 252-254 does not match. There seems something wrong to be happened.
- Please explain why the algorithm use k-4 in line 354.
- Figure 15 was not cited in the text and discussion or explanation of the result in Figure 15 should be included in the manuscript.
- In conclusions, the authors described that the second contribution of the manuscript is integrity assessment. However, any result on the integrity assessment could not found in the manuscript.
- Typo errors; line 125 EKF should be deleted., where line 130 and others should begin without indentation, in (10) daig-> diag, line 379 time-discrete -> descrete-time
Round 2
Reviewer 2 Report
I consider that most of my commentaries and suggestions were attended. However, there are still some points that demand a more detailed explanation.
About Q3, it is not so obvious to visualize how the green box in figure 1 is detailed in figure 6. I suggest including some graphical resources in order to delimit the same green area of figure 1 in figure 6.
About Q17, my priority suggestion is to include the reference used in the cover letter, even knowing that it is a self-citation. In this case, I understand that is perfectly reasonable. If the authors do not agree, I suggest including the same argumentation made on the cover letter.
About Q20, Figure 13 still has two letters (a), one on page 18, and another one on page 19.
Finally, I still insist on a better explanation about the Configuration Description (subsection 3.1). It is not easy to visualize the same reference for both systems. Maybe presenting a table with the features of both configurations would aid to compare them. Besides this, just saying that the reference system is ..." a trustable reference system" without any reference or data to validate it is not a good scientific practice. This aspect must be improved.
Reviewer 3 Report
Integrity assessment results would be presented in the experiment scenarios.
Only positioning improvement results do not seem to be enough to the readers.
